# Integration Methods and Optimization Algorithms

**Damien Scieur**
INRIA, ENS,
PSL Research University,
Paris France
damien.scieur@inria.fr

**Vincent Roulet**
INRIA, ENS,
PSL Research University,
Paris France
vincent.roulet@inria.fr

**Francis Bach**
INRIA, ENS,
PSL Research University,
Paris France
francis.bach@inria.fr

**Alexandre d'Aspremont**
CNRS, ENS
PSL Research University,
Paris France
aspremon@ens.fr

## Abstract

We show that accelerated optimization methods can be seen as particular instances of multi-step integration schemes from numerical analysis, applied to the gradient flow equation. Compared with recent advances in this vein, the differential equation considered here is the basic gradient flow, and we derive a class of multi-step schemes which includes accelerated algorithms, using classical conditions from numerical analysis. Multi-step schemes integrate the differential equation using larger step sizes, which intuitively explains the acceleration phenomenon.

## Introduction

Applying the gradient descent algorithm to minimize a function $f$ has a simple numerical interpretation as the integration of the gradient flow equation, written

$$\dot{x}(t) = -\nabla f(x(t)), \quad x(0) = x_0 \qquad \text{(Gradient Flow)}$$

using Euler's method. This appears to be a somewhat unique connection between optimization and numerical methods, since these two fields have inherently different goals. On one hand, numerical methods aim to get a precise discrete approximation of the solution $x(t)$ on a finite time interval. On the other hand, optimization algorithms seek to find the minimizer of a function, which corresponds to the infinite time horizon of the gradient flow equation. More sophisticated methods than Euler's were developed to get better consistency with the continuous time solution but still focus on a finite time horizon [see e.g. Süli and Mayers, 2003]. Similarly, structural assumptions on $f$ lead to more sophisticated optimization algorithms than the gradient method, such as the mirror gradient method [see e.g. Ben-Tal and Nemirovski, 2001; Beck and Teboulle, 2003], proximal gradient method [Nesterov, 2007] or a combination thereof [Duchi et al., 2010; Nesterov, 2015]. Among them Nesterov's accelerated gradient algorithm [Nesterov, 1983] is proven to be optimal on the class of smooth convex or strongly convex functions. This latter method was designed with optimal complexity in mind, but the proof relies on purely algebraic arguments and the key mechanism behind acceleration remains elusive, with various interpretations discussed in e.g. [Bubeck et al., 2015; Allen Zhu and Orecchia, 2017; Lessard et al., 2016].

Another recent stream of papers used differential equations to model the acceleration behavior and offer another interpretation of Nesterov's algorithm [Su et al., 2014; Krichene et al., 2015; Wibisono et al., 2016; Wilson et al., 2016]. However, the differential equation is often quite complex, being

reverse-engineered from Nesterov's method itself, thus losing the intuition. Moreover, integration methods for these differential equations are often ignored or are not derived from standard numerical integration schemes.

Here, we take another approach. Rather than using a complicated differential equation, we use advanced multistep discretization methods on the basic gradient flow equation in (Gradient Flow). Ensuring that the methods effectively integrate this equation for infinitesimal step sizes is essential for the continuous time interpretation and leads to a family of integration methods which contains various well-known optimization algorithms. A full analysis is carried out for linear gradient flows (quadratic optimization) and provides compelling explanations for the acceleration phenomenon. In particular, Nesterov's method can be seen as a stable and consistent gradient flow discretization scheme that allows bigger step sizes in integration, leading to faster convergence.

# 1   Gradient flow

We seek to minimize a $L$-smooth $\mu$-strongly convex function defined on $\mathbb{R}^d$. We discretize the gradient flow equation (Gradient Flow), given by the following ordinary differential equation

$$\dot{x}(t) = g(x(t)), \quad x(0) = x_0, \qquad \text{(ODE)}$$

where $g$ comes from a potential $-f$, meaning $g = -\nabla f$. Smoothness of $f$ means Lipschitz continuity of $g$, i.e.

$$\|g(x) - g(y)\| \le L\|x - y\|, \quad \text{for every } x, y \in \mathbb{R}^d,$$

where $\|.\|$ is the Euclidean norm. This property ensures existence and uniqueness of the solution of (ODE) (see [Süli and Mayers, 2003, Theorem 12.1]). Strong convexity of $f$ also means strong monotonicity of $-g$, i.e.,

$$\mu\|x - y\|^2 \le -\langle x - y, g(x) - g(y)\rangle, \quad \text{for every } x, y \in \mathbb{R}^d,$$

and ensures that (ODE) has a unique point $x^*$ such that $g(x^*) = 0$, called the equilibrium. This is the minimizer of $f$ and the limit point of the solution, i.e. $x(\infty) = x^*$. Finally this assumption allows us to control the convergence rate of the potential $f$ and the solution $x(t)$ as follows.

**Proposition 1.1** *Let $f$ be a $L$-smooth and $\mu$-strongly convex function and $x_0 \in \mathbf{dom}(f)$. Writing $x^*$ the minimizer of $f$, the solution $x(t)$ of (Gradient Flow) satisfies*

$$f(x(t)) - f(x^*) \le (f(x_0) - f(x^*))e^{-2\mu t}, \qquad \|x(t) - x^*\| \le \|x_0 - x^*\|e^{-\mu t}. \qquad (1)$$

A proof of this last result is recalled in the Supplementary Material. We now focus on numerical methods to integrate (ODE).

# 2   Numerical integration of differential equations

## 2.1   Discretization schemes

In general, we do not have access to an explicit solution $x(t)$ of (ODE). We thus use integration algorithms to approximate the curve $(t, x(t))$ by a grid $(t_k, x_k) \approx (t_k, x(t_k))$ on a finite interval $[0, t_{\max}]$. For simplicity here, we assume the step size $h_k = t_k - t_{k-1}$ is constant, i.e., $h_k = h$ and $t_k = kh$. The goal is then to minimize the approximation error $\|x_k - x(t_k)\|$ for $k \in [0, t_{\max}/h]$. We first introduce Euler's method to illustrate this on a basic example.

**Euler's explicit method.**    Euler's (explicit) method is one of the oldest and simplest schemes for integrating the curve $x(t)$. The idea stems from a Taylor expansion of $x(t)$ which reads

$$x(t + h) = x(t) + h\dot{x}(t) + O(h^2).$$

When $t = kh$, Euler's method approximates $x(t + h)$ by $x_{k+1}$, neglecting the second order term,

$$x_{k+1} = x_k + hg(x_k).$$

In optimization terms, we recognize the gradient descent algorithm used to minimize $f$. Approximation errors in an integration method accumulate with iterations, and as Euler's method uses only the last point to compute the next one, it has only limited control over the accumulated error.

**Linear multistep methods.** Multi-step methods use a combination of past iterates to improve convergence. Throughout the paper, we focus on *linear $s$-step methods* whose recurrence can be written

$$x_{k+s} = -\sum_{i=0}^{s-1} \rho_i x_{k+i} + h \sum_{i=0}^{s} \sigma_i g(x_{k+i}), \quad \text{for } k \geq 0,$$

where $\rho_i, \sigma_i \in \mathbb{R}$ are the parameters of the multistep method and $h$ is again the step size. Each new point $x_{k+s}$ is a function of the information given by the $s$ previous points. If $\sigma_s = 0$, each new point is given explicitly by the $s$ previous points and the method is called **explicit**. Otherwise each new point requires solving an implicit equation and the method is called **implicit**.

To simplify notations we use the shift operator $E$, which maps $Ex_k \to x_{k+1}$. Moreover, if we write $g_k = g(x_k)$, then the shift operator also maps $Eg_k \to g_{k+1}$. Recall that a univariate polynomial is called monic if its leading coefficient is equal to 1. We now give the following concise definition of $s$-step linear methods.

**Definition 2.1** *Given an* (ODE) *defined by $g, x_0$, a step size $h$ and $x_1, \ldots, x_{s-1}$ initial points, a **linear $s$-step method** generates a sequence $(t_k, x_k)$ which satisfies*

$$\rho(E)x_k = h\sigma(E)g_k, \quad \text{for every } k \geq 0, \tag{2}$$

*where $\rho$ is a monic polynomial of degree $s$ with coefficients $\rho_i$, and $\sigma$ a polynomial of degree $s$ with coefficients $\sigma_i$.*

A linear $s-$step method is uniquely defined by the polynomials $(\rho, \sigma)$. The sequence generated by the method then depends on the initial points and the step size. We now recall a few results describing the performance of multistep methods.

## 2.2 Stability

Stability is a key concept for integration methods. First of all, consider two curves $x(t)$ and $y(t)$, both solutions of (ODE), but starting from different points $x(0)$ and $y(0)$. If the function $g$ is Lipchitz-continuous, it is possible to show that the distance between $x(t)$ and $y(t)$ is bounded on a finite interval, i.e.

$$\|x(t) - y(t)\| \leq C\|x(0) - y(0)\| \qquad \forall t \in [0, t_{\max}],$$

where $C$ may depend exponentially on $t_{\max}$. We would like to have a similar behavior for our sequences $x_k$ and $y_k$, approximating $x(t_k)$ and $y(t_k)$, i.e.

$$\|x_k - y_k\| \approx \|x(t_k) - y(t_k)\| \leq C\|x(0) - y(0)\| \qquad \forall k \in [0, t_{\max}/h], \tag{3}$$

when $h \to 0$, so $k \to \infty$. Two issues quickly arise. First, for a linear $s$-step method, we need $s$ starting values $x_0, \ldots, x_{s-1}$. Condition (3) will therefore depend on all these starting values and not only $x_0$. Secondly, any discretization scheme introduces at each step an approximation error, called local error, which accumulates over time. We write this error $\epsilon^{\mathrm{loc}}(x_{k+s})$ and define it as $\epsilon^{\mathrm{loc}}(x_{k+s}) \triangleq x_{k+s} - x(t_{k+s})$, where $x_{k+s}$ is computed using the real solution $x(t_k), \ldots, x(t_{k+s-1})$. In other words, the difference between $x_k$ and $y_k$ can be described as follows

$$\|x_k - y_k\| \leq \text{Error in the initial condition} + \text{Accumulation of local errors}.$$

We now write a complete definition of stability, inspired by Definition 6.3.1 from Gautschi [2011].

**Definition 2.2 (Stability)** *A linear multistep method is stable iff, for two sequences $x_k$, $y_k$ generated by $(\rho, \sigma)$ using a sufficiently small step size $h > 0$, from the starting values $x_0, \ldots, x_{s-1}$, and $y_0, \ldots, y_{s-1}$, we have*

$$\|x_k - y_k\| \leq C\Big( \max_{i \in \{0, \ldots, s-1\}} \|x_i - y_i\| + \sum_{i=1}^{t_{\max}/h} \|\epsilon^{loc}(x_{i+s})\| + \|\epsilon^{loc}(y_{i+s})\| \Big), \tag{4}$$

*for any $k \in [0, t_{\max}/h]$. Here, the constant $C$ may depend on $t_{\max}$ but is independent of $h$.*

When $h$ tends to zero, we may recover equation (3) only if the accumulated local error also tends to zero. We thus need

$$\lim_{h \to 0} \frac{1}{h} \|\epsilon^{\mathrm{loc}}(x_{i+s})\| = 0 \quad \forall i \in [0, t_{\max}/h].$$

This condition is called *consistency*. The following proposition shows there exist simple conditions to check consistency, which rely on comparing a Taylor expansion of the solution with the coefficients of the method. Its proof and further details are given in the Supplementary Material.

**Proposition 2.3 (Consistency)** *A linear multistep method defined by polynomials* $(\rho, \sigma)$ *is consistent if and only if*

$$\rho(1) = 0 \qquad and \qquad \rho'(1) = \sigma(1). \tag{5}$$

Assuming consistency, we still need to control sensitivity to initial conditions, written

$$\|x_k - y_k\| \leq C \max_{i \in \{0, \dots, s-1\}} \|x_i - y_i\|. \tag{6}$$

Interestingly, analyzing the special case where $g = 0$ is completely equivalent to the general case and this condition is therefore called *zero-stability*. This reduces to standard linear algebra results as we only need to look at the solution of the homogeneous difference equation $\rho(E)x_k = 0$. This is captured in the following theorem whose technical proof can be found in [Gautschi, 2011, Theorem 6.3.4].

**Theorem 2.4 (Root condition)** *Consider a linear multistep method* $(\rho, \sigma)$. *The method is zero-stable if and only if all roots of* $\rho(z)$ *are in the unit disk, and the roots on the unit circle are simple.*

## 2.3 Convergence of the global error

Numerical analysis focuses on integrating an ODE on a finite interval of time $[0, t_{\max}]$. It studies the behavior of the global error defined by $x(t_k) - x_k$, as a function of the step size $h$. If the global error converges to $0$ with the step size, the method is guaranteed to approximate correctly the ODE on the time interval, for $h$ small enough.

We now state *Dahlquist's equivalence theorem*, which shows that the global error converges to zero when $h$ does if the method is *stable*, i.e. when the method is *consistent* and *zero-stable*. This naturally needs the additional assumption that the starting values $x_0, \dots, x_{s-1}$ are computed such that they converge to the solution $(x(0), \dots, x(t_{s-1}))$. The proof of the theorem can be found in Gautschi [2011].

**Theorem 2.5 (Dahlquist's equivalence theorem)** *Given an* (ODE) *defined by* $g$ *and* $x_0$ *and a consistent linear multistep method* $(\rho, \sigma)$, *whose starting values are computed such that* $\lim_{h \to 0} x_i = x(t_i)$ *for any* $i \in \{0, \dots, s-1\}$, *zero-stability is necessary and sufficient for convergence, i.e. to ensure* $x(t_k) - x_k \to 0$ *for any* $k$ *when the step size* $h$ *goes to zero.*

## 2.4 Region of absolute stability

The results above ensure stability and global error bounds on finite time intervals. Solving optimization problems however requires looking at infinite time horizons. We start by finding conditions ensuring that the numerical solution does not diverge when the time interval increases, i.e. that the numerical solution is stable with a constant $C$ which *does not depend on* $t_{\max}$. Formally, for a fixed step-size $h$, we want to ensure

$$\|x_k\| \leq C \max_{i \in \{0, \dots, s-1\}} \|x_i\| \quad \text{for all } k \in [0, t_{\max}/h] \text{ and } t_{\max} > 0. \tag{7}$$

This is not possible without further assumptions on the function $g$ as in the general case the solution $x(t)$ itself may diverge. We begin with the simple scalar linear case which, given $\lambda > 0$, reads

$$\dot{x}(t) = -\lambda x(t), \quad x(0) = x_0. \tag{Scalar Linear ODE}$$

The recurrence of a linear multistep methods with parameters $(\rho, \sigma)$ applied to (Scalar Linear ODE) then reads

$$\rho(E)x_k = -\lambda h \sigma(E)x_k \quad \Leftrightarrow \quad [\rho + \lambda h \sigma](E)x_k = 0,$$

where we recognize a homogeneous recurrence equation. Condition (7) is then controlled by the step size $h$ and the constant $\lambda$, ensuring that this homogeneous recurrent equation produces bounded solutions. This leads us to the definition of the region of absolute stability, also called A-stability.

**Definition 2.6 (Absolute stability)** *The region of absolute stability of a linear multistep method defined by $(\rho, \sigma)$ is the set of values $\lambda h$ such that the characteristic polynomial*

$$\pi_{\lambda h}(z) \triangleq \rho(z) + \lambda h\, \sigma(z) \tag{8}$$

*of the homogeneous recurrence equation $\pi_{\lambda h}(E)x_k = 0$ produces bounded solutions.*

Standard linear algebra links this condition to the roots of the characteristic polynomial as recalled in the next proposition (see e.g. Lemma 12.1 of Süli and Mayers [2003]).

**Proposition 2.7** *Let $\pi$ be a polynomial and write $x_k$ a solution of the homogeneous recurrence equation $\pi(E)x_k = 0$ with arbitrary initial values. If all roots of $\pi$ are inside the unit disk and the ones on the unit circle have a multiplicity exactly equal to one, then $\|x_k\| \leq \infty$.*

Absolute stability of a linear multistep method determines its ability to integrate a linear ODE defined by

$$\dot{x}(t) = -Ax(t), \quad x(0) = x_0, \tag{Linear ODE}$$

where $A$ is a positive symmetric matrix whose eigenvalues belong to $[\mu, L]$ for $0 < \mu \leq L$. In this case the step size $h$ must indeed be chosen such that for any $\lambda \in [\mu, L]$, $\lambda h$ belongs to the region of absolute stability of the method. This (Linear ODE) is a special instance of (Gradient Flow) where $f$ is a quadratic function. Therefore absolute stability gives a necessary (but not sufficient) condition to integrate (Gradient Flow) on $L$-smooth, $\mu$-strongly convex functions.

## 2.5 Convergence analysis in the linear case

By construction, absolute stability also gives hints on the convergence of $x_k$ to the equilibrium in the linear case. More precisely, it allows us to control the rate of convergence of $x_k$, approximating the solution $x(t)$ of (Linear ODE) as shown in the following proposition whose proof can be found in Supplementary Material.

**Proposition 2.8** *Given a (Linear ODE) defined by $x_0$ and a positive symmetric matrix $A$ whose eigenvalues belong to $[\mu, L]$ with $0 < \mu \leq L$, using a linear multistep method defined by $(\rho, \sigma)$ and applying a fixed step size $h$, we define $r_{\max}$ as*

$$r_{\max} = \max_{\lambda \in [\mu, L]} \ \max_{r \in \mathrm{roots}(\pi_{\lambda h}(z))} |r|,$$

*where $\pi_{\lambda h}$ is defined in (8). If $r_{\max} < 1$ and its multiplicity is equal to $m$, then the speed of convergence of the sequence $x_k$ produced by the linear multistep method to the equilibrium $x^*$ of the differential equation is given by*

$$\|x_k - x^*\| = O(k^{m-1} r_{\max}^k). \tag{9}$$

We can now use these properties to analyze and design multistep methods.

## 3 Analysis and design of multi-step methods

As shown previously, we want to integrate (Gradient Flow) and Proposition 1.1 gives a rate of convergence in the continuous case. If the method tracks $x(t)$ with sufficient accuracy, then the rate of the method will be close to the rate of convergence of $x(kh)$. So, *larger values of $h$ yield faster convergence of $x(t)$ to the equilibrium $x^*$.* However $h$ cannot be too large, as the method may be too inaccurate and/or unstable as $h$ increases. *Convergence rates of optimization algorithms are thus controlled by our ability to discretize the gradient flow equation using large step sizes.* We recall the different conditions that proper linear multistep methods should satisfy.

- *Monic polynomial (Section 2.1).* Without loss of generality (dividing both sides of the difference equation of the multistep method (2) by $\rho_s$ does not change the method).
- *Explicit method (Section 2.1).* We assume that the scheme is explicit in order to avoid solving a non-linear system at each step.

- *Consistency (Section 2.2).* If the method is not consistent, then the local error does not converge when the step size goes to zero.
- *Zero-stability (Section 2.2).* Zero-stability ensures convergence of the global error (Section 2.3) when the method is also consistent.
- *Region of absolute stability (Section 2.4).* If $\lambda h$ is not inside the region of absolute stability for any $\lambda \in [\mu, L]$, then the method is divergent when $t_{\max}$ increases.

Using the remaining degrees of freedom, we can tune the algorithm to improve the convergence rate on (Linear ODE), which corresponds to the optimization of a quadratic function. Indeed, as showed in Proposition 2.8, the largest root of $\pi_{\lambda h}(z)$ gives us the rate of convergence on quadratic functions (when $\lambda \in [\mu, L]$). Since smooth and strongly convex functions are close to quadratic (being sandwiched between two quadratics), this will also give us a good idea of the rate of convergence on these functions. We do not derive a proof of convergence of the sequence for general smooth and (strongly) convex function (but convergence is proved by Nesterov [2013] or using Lyapunov techniques by Wilson et al. [2016]). Still our results provide intuition on *why* accelerated methods converge faster.

## 3.1 Analysis of two-step methods

We now analyze convergence of two-step methods (an analysis of Euler's method is provided in the Supplementary Material). We first translate the conditions multistep method, listed at the beginning of this section, into constraints on the coefficients:

$$
\begin{aligned}
\rho_2 &= 1 && \text{(Monic polynomial)} \\
\sigma_2 &= 0 && \text{(Explicit method)} \\
\rho_0 + \rho_1 + \rho_2 &= 0 && \text{(Consistency)} \\
\sigma_0 + \sigma_1 + \sigma_2 &= \rho_1 + 2\rho_2 && \text{(Consistency)} \\
|\text{Roots}(\rho)| &\leq 1 && \text{(Zero-stability)}.
\end{aligned}
$$

Equality contraints yield three linear constraints, defining the set $\mathcal{L}$ such that

$$
\mathcal{L} = \{\rho_0, \rho_1, \sigma_0, \sigma_1 : \quad \rho_1 = -(1 + \rho_0), \quad \sigma_1 = 1 - \rho_0 - \sigma_0, \quad |\rho_0| < 1\}. \tag{10}
$$

We now seek conditions on the remaining parameters to produce a stable method. Absolute stability requires that all roots of the polynomial $\pi_{\lambda h}(z)$ in (8) are inside the unit circle, which translates into condition on the roots of second order equations here. The following proposition gives the values of the roots of $\pi_{\lambda h}(z)$ as a function of the parameters $\rho_i$ and $\sigma_i$.

**Proposition 3.1** *Given constants $0 < \mu \leq L$, a step size $h > 0$ and a linear two-step method defined by $(\rho, \sigma)$, under the conditions*

$$
(\rho_1 + \mu h \sigma_1)^2 \leq 4(\rho_0 + \mu h \sigma_0), \qquad (\rho_1 + L h \sigma_1)^2 \leq 4(\rho_0 + L h \sigma_0), \tag{11}
$$

*the roots $r_\pm(\lambda)$ of $\pi_{\lambda h}$, defined in (8), are complex conjugate for any $\lambda \in [\mu, L]$. Moreover, the largest root modulus is equal to*

$$
\max_{\lambda \in [\mu, L]} |r_\pm(\lambda)|^2 = \max\{\rho_0 + \mu h \sigma_0, \ \rho_0 + L h \sigma_0\}. \tag{12}
$$

The proof can be found in the Supplementary Material. The next step is to minimize the largest modulus (12) in the coefficients $\rho_i$ and $\sigma_i$ to get the best rate of convergence, assuming the roots are complex (the case were the roots are real leads to weaker results).

## 3.2 Design of a family of two-step methods for quadratics

We now have all ingredients to build a two-step method for which the sequence $x_k$ converges quickly to $x^*$ for quadratic functions. Optimizing the convergence rate means solving the following problem,

$$
\begin{aligned}
\min \quad & \max\{\rho_0 + \mu h \sigma_0, \ \rho_0 + L h \sigma_0\} \\
\text{s.t.} \quad & (\rho_0, \rho_1, \sigma_0, \sigma_1) \in \mathcal{L} \\
& (\rho_1 + \mu h \sigma_1)^2 \leq 4(\rho_0 + \mu h \sigma_0) \\
& (\rho_1 + L h \sigma_1)^2 \leq 4(\rho_0 + L h \sigma_0),
\end{aligned}
$$

in the variables $\rho_0, \rho_1, \sigma_0, \sigma_1, h > 0$, where $\mathcal{L}$ is defined in (10). If we use the equality constraints in (10) and make the following change of variables,

$$\hat{h} = h(1 - \rho_0), \quad c_\mu = \rho_0 + \mu h \sigma_0, \quad c_L = \rho_0 + L h \sigma_0, \tag{13}$$

the problem can be solved, for fixed $\hat{h}$, in the variables $c_\mu, c_L$. In that case, the optimal solution is given by

$$c_\mu^* = (1 - \sqrt{\mu\hat{h}})^2, \quad c_L^* = (1 - \sqrt{L\hat{h}})^2, \tag{14}$$

obtained by tightening the two first inequalities, for $\hat{h} \in\, ]0, \frac{(1+\mu/L)^2}{L}[$. Now if we fix $\hat{h}$ we can recover a two step linear method defined by $(\rho, \sigma)$ and a step size $h$ by using the equations in (13). We define the following quantity $\beta = (1 - \sqrt{\mu/L})/(1 + \sqrt{\mu/L})$.

**A suboptimal two-step method.** Setting $\hat{h} = 1/L$ for example, the parameters of the corresponding two-step method, called method $\mathcal{M}_1$, are

$$\mathcal{M}_1 = \left\{ \rho(z) = \beta - (1 + \beta)z + z^2, \quad \sigma(z) = -\beta(1 - \beta) + (1 - \beta^2)z, \quad h = \frac{1}{L(1 - \beta)} \right\} \tag{15}$$

and its largest modulus root (12) is given by

$$\text{rate}(\mathcal{M}_1) = \sqrt{\max\{c_\mu,\ c_L\}} = \sqrt{c_\mu} = 1 - \sqrt{\mu/L}.$$

**Optimal two-step method for quadratics.** We can compute the optimal $\hat{h}$ which minimizes the maximum of the two roots $c_\mu^*$ and $c_L^*$ defined in (14). The solution simply balances the two terms in the maximum, with $\hat{h}^* = (1 + \beta)^2/L$. This choice of $\hat{h}$ leads to the method $\mathcal{M}_2$, described by

$$\mathcal{M}_2 = \left\{ \rho(z) = \beta^2 - (1 + \beta^2)z + z^2, \quad \sigma(z) = (1 - \beta^2)z, \quad h = \frac{1}{\sqrt{\mu L}} \right\} \tag{16}$$

with convergence rate

$$\text{rate}(\mathcal{M}_2) = \sqrt{c_\mu} = \sqrt{c_L} = \beta = (1 - \sqrt{\mu/L})/(1 + \sqrt{\mu/L}) < \text{rate}(\mathcal{M}_1).$$

We will now see that methods $\mathcal{M}_1$ and $\mathcal{M}_2$ are actually related to Nesterov's accelerated method and Polyak's heavy ball algorithms.

## 4 On the link between integration and optimization

In the previous section, we derived a family of linear multistep methods, parametrized by $\hat{h}$. We will now compare these methods to common optimization algorithms used to minimize $L$-smooth, $\mu$-strongly convex functions.

### 4.1 Polyak's heavy ball method

The heavy ball method was proposed by Polyak [1964]. It adds a momentum term to the gradient step

$$x_{k+2} = x_{k+1} - c_1 \nabla f(x_{k+1}) + c_2(x_{k+1} - x_k),$$

where $c_1 = 4/(\sqrt{L} + \sqrt{\mu})^2$ and $c_2 = \beta^2$. We can organize the terms in the sequence to match the general structure of linear multistep methods, to get

$$\beta^2 x_k - (1 + \beta^2)x_{k+1} + x_{k+2} = c_1\left(-\nabla f(x_{k+1})\right).$$

We easily identify $\rho(z) = \beta^2 - (1+\beta^2)z + z^2$ and $h\sigma(z) = c_1 z$. To extract $h$, we will assume that the method is consistent (see conditions (5)). All computations done, we can identify the corresponding linear multistep method as

$$\mathcal{M}_{\text{Polyak}} = \left\{ \rho(z) = \beta^2 - (1 + \beta^2)z + 1, \quad \sigma(z) = (1 - \beta^2)z, \quad h = \frac{1}{\sqrt{\mu L}} \right\}. \tag{17}$$

This shows that $\mathcal{M}_{\text{Polyak}} = \mathcal{M}_2$. In fact, this result was expected since Polyak's method is known to be optimal for quadratic functions. However, it is also known that Polyak's algorithm does not converge for a general smooth and strongly convex function [Lessard et al., 2016].

## 4.2 Nesterov's accelerated gradient

Nesterov's accelerated method in its simplest form is described by two sequences $x_k$ and $y_k$, with

$$
\begin{aligned}
y_{k+1} &= x_k - \frac{1}{L}\nabla f(x_k), \\
x_{k+1} &= y_{k+1} + \beta(y_{k+1} - y_k).
\end{aligned}
$$

As above, we will write Nesterov's accelerated gradient as a linear multistep method by expanding $y_k$ in the definition of $x_k$, to get

$$
\beta x_k - (1+\beta)x_{k+1} + x_{k+2} = \frac{1}{L}\left(-\beta(-\nabla f(x_k)) + (1+\beta)(-\nabla f(x_{k+1}))\right).
$$

Again, assuming as above that the method is consistent to extract $h$, we identify the linear multistep method associated to Nesterov's algorithm. After identification,

$$
\mathcal{M}_{\text{Nest}} = \left\{ \rho(z) = \beta - (1+\beta)z + z^2, \quad \sigma(z) = -\beta(1-\beta) + (1-\beta^2)z, \quad h = \frac{1}{L(1-\beta)}, \right\}
$$

which means that $\mathcal{M}_1 = \mathcal{M}_{\text{Nest}}$.

## 4.3 The convergence rate of Nesterov's method

Pushing the analysis a little bit further, we have a simple intuitive argument that explains *why* Nesterov's algorithm is faster than the gradient method. There is of course a complete proof of its rate of convergence [Nesterov, 2013], even using differential equations arguments [Wibisono et al., 2016; Wilson et al., 2016], but we take a simpler approach here. The key parameter is the step size $h$. If we compare it with the step size in the classical gradient method, Nesterov's method uses a step size which is $(1-\beta)^{-1} \approx \sqrt{L/\mu}$ times larger.

Recall that, in continuous time, the rate of convergence of $x(t)$ to $x^*$ is given by

$$
f(x(t)) - f(x^*) \leq e^{-2\mu t}(f(x_0) - f(x^*)).
$$

The gradient method tries to approximate $x(t)$ using an Euler scheme with step size $h = 1/L$, which means $x_k^{(\text{grad})} \approx x(k/L)$, so

$$
f(x_k^{(\text{grad})}) - f(x^*) \approx f(x(k/L)) - f(x^*) \leq (f(x_0) - f(x^*))e^{-2k\frac{\mu}{L}}.
$$

However, Nesterov's method has a step size equal to

$$
h_{\text{Nest}} = \frac{1}{L(1-\beta)} = \frac{1+\sqrt{\mu/L}}{2\sqrt{\mu L}} \approx \frac{1}{\sqrt{4\mu L}} \qquad \text{which means} \quad x_k^{\text{nest}} \approx x\left(k/\sqrt{4\mu L}\right).
$$

while maintaining stability. In that case, the estimated rate of convergence becomes

$$
f(x_k^{\text{nest}}) - f(x^*) \approx f\left(x\left(k/\sqrt{4\mu L}\right)\right) - f(x^*) \leq (f(x_0) - f(x^*))e^{-k\sqrt{\mu/L}},
$$

which is approximatively the rate of convergence of Nesterov's algorithm in discrete time and we recover the accelerated rate in $\sqrt{\mu/L}$ versus $\mu/L$ for gradient descent.

Overall, the accelerated method is more efficient because it integrates the gradient flow *faster* than simple gradient descent, making longer steps. A numerical simulation in Figure 1 makes this argument more visual.

## 5 Generalization and Future Work

We showed that accelerated optimization methods can be seen as multistep integration schemes applied to the basic gradient flow equation. Our results give a natural interpretation of acceleration: multistep schemes allow for larger steps, which speeds up convergence. In the Supplementary Material, we detail further links between integration methods and other well-known optimization algorithms such as proximal point methods, mirror gradient decent, proximal gradient descent, and

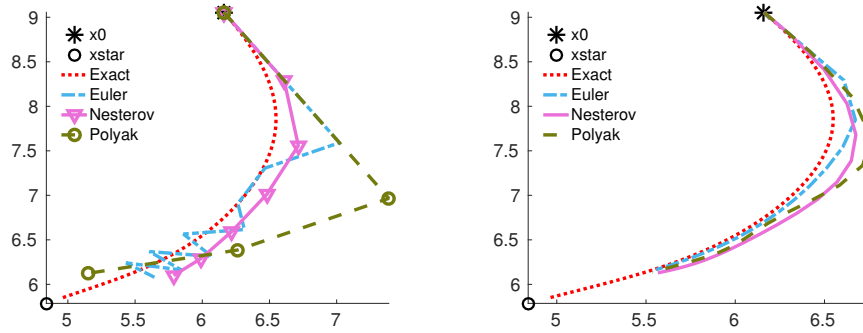

Figure 1: Integration of a linear ODE with optimal (left) and small (right) step sizes.

discuss the weakly convex case. The extra-gradient algorithm and its recent accelerated version Diakonikolas and Orecchia [2017] can also be linked to another family of integration methods called Runge-Kutta which include notably predictor-corrector methods.

Our stability analysis is limited to the quadratic case, the definition of A-stability being too restrictive for the class of smooth and strongly convex functions. A more appropriate condition would be G-stability, which extends A-stability to non-linear ODEs, but this condition requires strict monotonicity of the error (which is not the case with accelerated algorithms). Stability may also be tackled by recent advances in lower bound theory provided by Taylor [2017] but these yield numerical rather than analytical convergence bounds. Our next objective is thus to derive a new stability condition in between A-stability and G-stability.

## Acknowledgments

The authors would like to acknowledge support from a starting grant from the European Research Council (ERC project SIPA), from the European Union's Seventh Framework Programme (FP7-PEOPLE-2013-ITN) under grant agreement number 607290 SpaRTaN, an AMX fellowship, as well as support from the chaire *Économie des nouvelles données* with the *data science* joint research initiative with the *fonds AXA pour la recherche* and a gift from Société Générale Cross Asset Quantitative Research.

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
