[Supplementary Material]

# Integration Methods and Optimization Algorithms
# Supplementary Material

## Overview

This Supplementary Material is organized as follows. We start in Section 6 by detailing the consistency condition, its motivation, proof and interpretation. Then in Section 7 we analyze the integration properties of Euler's scheme, in particular its stability. We extend then the intuitive explanation for acceleration (longer steps of the integration method) to the weakly convex case in Section 8. Sections 9,10 and 11 explore further links between integration methods and optimization algorithms. Finally missing proofs can be found in Section 12.

## 6 Consistency

We defined quickly consistency by looking at the stability of an integration method, we detail here a formal definition. The truncation error of a linear multi-step method is a measure of the local error $\epsilon_{\text{loc}}(x_k)$ made by the method, normalized by $h$. More precisely it is defined using the difference between the step performed by the algorithm and the step which reaches exactly $x(t_{k+s})$, with

$$T(h) \triangleq \frac{x(t_{k+s}) - x_{k+s}}{h} \qquad \text{assuming } x_{k+i} = x(t_{k+i}), \ i = 0, \ldots, s-1. \qquad (18)$$

This definition does not depend on $k$ but on the recurrence of the linear $s$-step method and on the (ODE) defined by $g$ and $x_0$. We can use this truncation error to define formally consistency.

**Definition 6.1.** *An integration method for an* (ODE) *defined by* $g, x_0$ *is consistent if and only if, for any initial condition* $x_0$,

$$\lim_{h \to 0} \|T(h)\| = 0.$$

Proposition 2.3 gave then simple conditions to check consistency. We recall it here and give its proof.

**Proposition.** *A linear multi-step method defined by polynomials* $(\rho, \sigma)$ *is consistent if and only if*

$$\rho(1) = 0 \qquad \text{and} \qquad \rho'(1) = \sigma(1). \qquad (19)$$

*Proof.* Assume $t_k = kh$. If we expand $g(x(t_k))$ we have

$$g(x(t_k)) = g(x_0) + O(h).$$

If we do the same thing with $x(t_k)$, we have

$$x(t_k) = x_0 + kh\dot{x}(t_0) + O(h^2) = x_0 + khg(x_0) + O(h^2).$$

If we plug these results in the linear multi-step method,

$$
\begin{aligned}
T(h) &= \frac{1}{h}\left( x(t_{k+s}) + \sum_{i=0}^{s-1} \rho_i x(t_{k+i}) - h\sum_{i=0}^{s} \sigma_i g(x(t_{k+i})) \right) \\
&= \frac{1}{h}\rho(1)x_0 + (\rho'(1) - \sigma(1))g(x_0) + O(h).
\end{aligned}
$$

The limit is equal to zero if and only if we satisfy (19). ∎

Consistency is crucial for integration methods, we give some intuition about how important are conditions defined in (19).

**First condition, $\rho(1) = 0$.** If the condition is not satisfied, then the method exhibits an artificial gain or damping. Assume we start at some equilibrium $x^*$ of ODE (i.e. $\nabla f(x^*) = 0$), so $x_i = x^*$ for the first $s-1$ steps. The next iterate becomes

$$x_s = -\sum_{i=0}^{s-1} \rho_i x^* + h\sigma(E) \underbrace{g(x^*)}_{=0},$$

and if $1 + \sum_{i=0}^{s} \rho_i = \rho(1) \neq 0$, we have that the next iterate $x_s$ is different from $x^*$.

**Second condition, $\rho'(1) = \sigma(1)$.** If this relation is not satisfied, we actually are integrating another equation than (ODE). Assuming the first condition satisfied, 1 is a root of $\rho$. Consider then the factorization

$$\rho(z) = (z-1)\tilde{\rho}(z)$$

where $\tilde{\rho}$ is a polynomial of degree $s-1$, and $\rho'(1) = \tilde{\rho}(1)$. The linear multi-step method becomes

$$\tilde{\rho}(E)(y_{k+1} - y_k) = h\sigma(E)g(y_k).$$

If we sum up the above equation from the initial point, we get

$$\tilde{\rho}(E)(y_k - y_0) = \sigma(E)G_k,$$

where $G_k = \sum_{i=0}^{k} hg(y_i)$. If $h$ goes to zero, our iterates $y_k$ converge to some continuous curve $c(t)$, and $G_k \to \int_0^t g(c(\tau))\,d\tau$,

$$\sum_{i=0}^{s} \tilde{\rho}_i(c(t) - x(0)) = \sum_{i=0}^{s-1} \sigma_i \int_0^t g(c(\tau))\,d\tau.$$

If we take the derivative over time, we get

$$\tilde{\rho}(1)\dot{c}(t) = \sigma(1)g(c(t)) \quad \Leftrightarrow \quad \rho'(1)\dot{c}(t) = \sigma(1)g(c(t)).$$

which is different from the ODE we wanted to discretize, unless $\rho'(1) = \sigma(1)$

## 7 Analysis and design of Euler's method

In Section 2.1 we introduced Euler's method. In fact, we can view it as an explicit linear "multi-step" method with $s = 1$ defined by the polynomials

$$\rho(z) = -1 + z, \quad \sigma(z) = 1.$$

We can check easily that it is consistent (using Proposition 2.3) and zero-stable since $\rho(z)$ has only one root which lies on the unit circle (Theorems 2.4 and 2.5). We need to determine the region of absolute stability in order to have an idea about the maximum value that $h > 0$ can take before the method becomes unstable. Assume we want to integrate any $\mu$-strongly convex and $L$-smooth function $f$, with $0 \leq \mu < L$ with any starting value $x_0$. Then using Proposition 2.7, we need to find the set of value of $h$ such that the roots of the polynomial

$$\pi_{\lambda h}(z) = [\rho + \lambda h\sigma](z) = -1 + \lambda h + z, \quad \lambda \in [\mu, L]$$

are small. The unique root is $1 - \lambda h$ and we need to solve the following minimax problem

$$\min_h \max_{\lambda \in [\mu, L]} |1 - \lambda h|,$$

in the variable $h > 0$. The solution of this optimization problem is $h^* = \frac{2}{L+\mu}$, its optimal value is $(L - \mu)/(L + \mu)$ and its rate of convergence is then

$$\|x_k - x^*\| = O\left(\left(\frac{1 - \mu/L}{1 + \mu/L}\right)^k\right).$$

We recover the optimal step size and the rate of convergence of the gradient method for a general smooth and strongly convex function [Nesterov, 2013].

# 8 Acceleration of weakly convex functions

By matching the coefficients of Nesterov's method, we deduced the value of the step-size used for the integration of (Gradient Flow). Then, using the rate of convergence of $x(t)$ to $x^*$, we estimated the rate of convergence of Nesterov's method assuming $x_k \approx x(t_k)$. Here, we will do the same but without assuming strong convexity. However, the estimation of the rate of convergence in discrete time needs the one in continuous time, described by the following proposition whose proof can be found in Section 12.3.

**Proposition 8.1.** *Let $f$ be $L$-smooth convex function, $x^*$ one of its minimizers and $x(t)$ be the solution of (Gradient Flow). Then*

$$f(x(t)) - f(x^*) \leq \frac{\|x_0 - x^*\|^2}{t + (2/L)}. \tag{20}$$

Assume we use Euler's method with step size $h = \frac{1}{L}$, the estimated rate of convergence will be

$$f(x_k) - f(x^*) \approx f(x(kh)) - f(x^*) \leq \frac{L\|x_0 - x^*\|^2}{k+2},$$

which is close to the rate of convergence of the classical gradient method for convex function. Now, consider Nesterov's method for minimizing a smooth and convex function $f$:

$$y_{k+1} = x_k - \frac{1}{L}\nabla f(x_k)$$
$$x_{k+1} = -\beta_k x_k + (1 + \beta_k)x_{k+1},$$

where $\beta_k \approx \frac{k-2}{k+1}$. If we expand everything, we get after rearrangement,

$$\beta_k x_{k-1} - (1 + \beta_k)x_k + x_{k+1} = \frac{1}{L}\left(\beta_k(-\nabla f(x_{k-1})) - (1 + \beta_k)(-\nabla f(x_k))\right).$$

In other terms, we have an expression of the form $\rho_k(E)x_k = h_k\sigma_k(E)(-\nabla f(x_k))$. We can identify $h$ if we assume the method consistent, which means

$$\rho(1) = 0 \qquad \text{Always satisfied}$$
$$h_k\rho_k'(1) = h_k\sigma_k(1) \qquad \Rightarrow h_k = \frac{1}{L(1 - \beta_{k+1})} = \frac{(k+2)}{3L}.$$

We can estimate, using (20), the rate of convergence of Nesterov's method. Since $x_k \approx x(t_k)$,

$$x_k \approx x\left(\sum_{i=0}^{k} h_i\right) \approx x\left(\frac{k^2}{6L}\right).$$

In terms of convergence to the optimal value,

$$f(x_k) - f(x^*) \approx f(x(t_k) - f(x^*) \leq \frac{6L\|x_0 - x^*\|^2}{k^2 + 12},$$

which is close to the bound from Nesterov [2013]. Again, because the step-size of Nesterov's algorithm is larger (while keeping a stable sequence), we converge faster than the Euler's method.

# 9 Proximal algorithms and implicit integration methods

We present here links between proximal algorithms and implicit numerical methods that integrate the gradient flow equation. We begin with Euler's implicit method that corresponds to the proximal point algorithm.

## 9.1 Euler's implicit method and proximal point algorithm

We saw in Section 2.1 that Euler's explicit method used the Taylor expansion of the solution $x(t)$ of the (ODE) at the current point. The implicit version uses the Taylor expansion at the next point which reads

$$x(t) = x(t + h) - h\dot{x}(t + h) + O(h^2).$$

If $t = kh$, by neglecting the second order term we get implicit Euler's method,

$$x_{k+1} = x_k + hg(x_{k+1}). \tag{21}$$

This recurrent equation requires to solve an implicit equation at each step that may be costly. However it provides better stability than the explicit version. This is generally the case for implicit methods (see Süli and Mayers [2003] for further details on implicit methods).

Now assume that $g$ comes from a potential $-f$ such that we are integrating (Gradient Flow). Solving the implicit equation (21) is equivalent to compute the proximal operator of $f$ defined as

$$\mathbf{prox}_{f,h}(x) = \underset{z}{\operatorname{argmin}} \frac{1}{2}\|z - x\|_2^2 + hf(z). \tag{22}$$

This can be easily verified by checking the first-order optimality conditions of the minimization problem. Euler's implicit method applied to (Gradient Flow) reads then

$$x_{k+1} = \mathbf{prox}_{f,h}(x_k),$$

where we recognize the proximal point algorithm [Rockafellar, 1976].

We present now Mixed ODE that corresponds to composite optimization problems.

## 9.2 Implicit Explicit methods and proximal gradient descent

In numerical analysis, it is common to consider the differential equation

$$\dot{x} = g(x) + \omega(x), \tag{Mixed ODE}$$

where $g(x)$ is considered as the "non-stiff" part of the problem and $\omega$ the stiff one, where stiffness may be assimilated to bad conditioning [Ascher et al., 1995; Frank et al., 1997]. Usually, we assume $\omega$ integrable using an implicit method. If $\omega$ derives from a potential $-\Omega$ (meaning $\omega = -\nabla\Omega$), this is equivalent to assume that the proximal operator of $\Omega$ defined in (22) can be computed exactly.

We approximate the solution of (Mixed ODE) using IMplicit-EXplicit schemes (IMEX). In our case, we will focus on the following multi-step based IMEX scheme,

$$\rho(E)x_k = h\big(\sigma(E)g(x_k) + \gamma(E)\omega(x_k)\big),$$

where $\rho, \sigma$ and $\gamma$ are polynomials of degrees $s, s-1$ (the explicit part) and $s$ respectively and $\rho$ is monic. It means that, at each iteration, we need to solve, in $x_{k+s}$,

$$x_{k+s} = \sum_{i=0}^{s-1} \underbrace{\big(-\rho_i x_{k+i} + \sigma_i hg(x_{k+i}) + \gamma_i h\omega(x_{k+i})\big)}_{\text{known}} + \gamma_s \omega(x_{k+s}).$$

In terms of optimization the mixed ODE corresponds to composite minimization problems of the form

$$\text{minimize} \quad f(x) + \Omega(x), \tag{23}$$

where $f, \Omega$ are convex and $\Omega$ has a computable proximal operator. We can link IMEX schemes with many optimization algorithms which use the proximal operator, such as proximal gradient method, FISTA or Nesterov's method. For example, proximal gradient is written

$$
\begin{aligned}
y_{k+1} &= x_k - h\nabla f(x_k) \\
x_{k+1} &= \mathbf{prox}_{h\Omega}(y_{k+1}).
\end{aligned}
$$

After expansion, we get

$$x_{k+1} = y_{k+1} - h\nabla\Omega(x_{k+1}) = x_k + hg(x_k) + h\omega(x_{k+1}),$$

which corresponds to the IMEX method with polynomials

$$\rho(z) = -1 + z, \quad \sigma(z) = 1, \quad \gamma(z) = z.$$

However, for Fista and Nesterov's method, we need to use a variant of linear multi-step algorithms, called *one leg* methods [Dahlquist, 1983; Zhang and Xiao, 2016]. Instead of combining the gradients, the idea is to compute $g$ at a linear combination of the previous points, i.e.

$$\rho(E)x_k = h\left(g(\sigma(E)x_k) + \omega(\gamma(E)x_k)\right).$$

Their analysis (convergence, consistency, interpretation of $h$, etc...) is slightly different from linear multi-step method, so we will not go into details in this paper, but the correspondence still holds.

## 9.3 Non-smooth gradient flow

In the last subsection we assumed that $\omega$ comes from a potential. However in the optimization literature, composite problems have a smooth convex part and a non-smooth sub-differentiable convex part which prevents us from interpreting the problem with the gradient flow ODE. Non-smooth convex optimization problems can be treated with differential inclusions (see [Bolte et al., 2007] for recent results on it)

$$\dot{x}(t) + \partial f(x(t)) \ni 0,$$

where $f$ is a sub-differentiable function whose sub-differential at $x$ is written $\partial f(x)$. Composite problems (23) can then be seen as the discretization of the differential inclusion

$$\dot{x}(t) + \nabla f(x(t)) + \partial \Omega x(t) \ni 0.$$

## 10 Mirror gradient descent and non-Euclidean gradient flow

In many optimization problems, it is common to replace the Euclidean geometry with a distance-generating function called $d(x)$, with the associated Bregman divergence

$$\mathcal{B}_d(x, y) = d(x) - d(y) - \langle \nabla d(y), x - y \rangle,$$

with $d$ strongly-convex and lower semi-continuous. To take into account this geometry we consider the Non-Euclidean Gradient Flow [Krichene et al., 2015]

$$\begin{aligned} \dot{y}(t) &= -\nabla f\left(x(t)\right) \\ x(t) &= \nabla d^*(y(t)) \\ x(0) &= x_0, \ y(0) = \nabla d(x_0). \end{aligned} \tag{NEGF}$$

Here $\nabla d$ maps primal variables to dual ones and, as $d$ is strongly convex, $(\nabla d)^{-1} = \nabla d^*$, where $d^*$ is the Fenchel conjugate of $d$. In fact, we can write (NEGF) using only one variable $y$, but this formulation has the advantage to exhibit both primal and dual variables $x(t)$ and $y(t)$. Applying the forward Euler's explicit method we get the following recurrent equation

$$y_{k+1} - y_k = -h \nabla f(x_k), \quad x_{k+1} = \nabla d^* y_{k+1}.$$

Now consider the mirror gradient scheme :

$$x_{k+1} = \underset{x}{\operatorname{argmin}} \ h \langle \nabla f(x_k), x \rangle + \mathcal{B}_h(x, x_k).$$

First optimality condition reads

$$\nabla_x \left( h \langle \nabla f(x_k), x \rangle + \mathcal{B}_h(x, x_k) \right) \big|_{x=x_{k+1}} = h \nabla f(x_k) + \nabla d(x_{k+1}) - \nabla d(x_k) = 0$$

Using that $(\nabla d)^{-1} = \nabla d^*$ we get

$$h \nabla f(x_k) + y_{k+1} - y_k = 0, \quad x_{k+1} = \nabla d^* y_{k+1},$$

which is exactly Euler's explicit method defined in (NEGF).

## 11 Universal gradient descent and generalized gradient flow

Consider the Generalized Gradient Flow, which combines the ideas of (Mixed ODE) and (NEGF),

$$\begin{aligned} \dot{y}(t) &= -\nabla f(x(t)) - \nabla \Omega(x(t)) \\ x(t) &= \nabla d^*(y(t)) \\ x(0) &= x_0, \ y(0) = \nabla d(x_0). \end{aligned} \tag{GGF}$$

We can write its ODE counterpart, called the "Generalized ODE",

$$\begin{aligned} \dot{y}(t) &= g(x(t)) + \omega(x(t)) \\ x(t) &= \nabla d^*(y(t)), \\ x(0) &= x_0, \ y(0) = \nabla d(x_0). \end{aligned} \tag{GODE}$$

where $g = -\nabla f$, with $f$ a smooth convex function, $d$ a strongly convex and semi-continuous distance generating function and $\omega = -\nabla\Omega$, where $\Omega$ is a simple convex function. If $\Omega$ is not differentiable we can consider the corresponding differential inclusion as presented in Section 9.3. Here we focus on (GODE) and (GGF) to highlight the links with integration methods. The discretization of this ODE is able to generate many algorithms in many different settings. For example, consider the IMEX explicit-implicit Euler's method,

$$\frac{y_{k+1} - y_k}{h} = g(x_k) + \omega(\nabla d^*(y_{k+1})), \quad x_{k+1} = \nabla d^*(y_{k+1}),$$

which can be decomposed into three steps,

$$
\begin{aligned}
z_{k+1} &= y_k + hg(x_k) & \text{(Gradient step in dual space)}, \\
y_{k+1} &= \mathbf{prox}_{h(\Omega\circ\nabla d^*)}(z_{k+1}) & \text{(Projection step in dual space)}, \\
x_{k+1} &= \nabla d^*(y_{k+1}) & \text{(Mapping back in primal space)}.
\end{aligned} \tag{24}
$$

Now consider the universal gradient method scheme presented by Nesterov [2015]:

$$x_{k+1} = \arg\min_x \langle \nabla f(x_k), x - x_k \rangle + \Omega(x) + \mathcal{B}_d(x, x_k).$$

Again we can show that both recursions are the same: if we write the first optimality condition,

$$
\begin{aligned}
0 &= \nabla_x \left( h\langle \nabla f(x_k), x - x_k \rangle + h\Omega(x) + \mathcal{B}(x, x_k) \right)\big|_{x=x_{k+1}} \\
&= hg(x_k) + h\partial\Omega(x_{k+1}) + \nabla d(x_{k+1}) - \nabla d(x_k) \\
&= \underbrace{hg(x_k) - y_k}_{=z_{k+1}} + h\partial\Omega(\nabla d^*(y_{k+1})) - y_{k+1}.
\end{aligned}
$$

We thus need to solve the non-linear system of equations

$$y_{k+1} = z_{k+1} + h\partial\Omega(\nabla d^*(y_{k+1})),$$

which is equivalent to the projection step (24). Then we simply recover $x_{k+1}$ by applying $\nabla d^*$ on $y_{k+1}$.

## 12 Missing Proofs

### 12.1 Proof of Proposition 1.1

**Proposition.** *Let $f$ be a $L$-smooth and $\mu$-strongly convex function and $x_0 \in \mathbf{dom}(f)$. Writing $x^*$ the minimizer of $f$, the solution $x(t)$ of (Gradient Flow) satisfies*

$$
\begin{aligned}
f(x(t)) - f(x^*) &\leq (f(x_0) - f(x^*))e^{-2\mu t} \tag{25} \\
\|x(t) - x^*\| &\leq \|x_0 - x^*\|e^{-\mu t}. \tag{26}
\end{aligned}
$$

*Proof.* Indeed, if we derive the left-hand-side of (25),

$$\frac{\mathrm{d}}{\mathrm{d}t}[f(x(t)) - f(x^*)] = \langle \nabla f(x(t)), \dot{x}(t) \rangle = -\|f'(x(t))\|^2.$$

Using that $f$ is strongly convex, we have (see Nesterov [2013])

$$f(x) - f(x^*) \leq \frac{1}{2\mu}\|\nabla f(x)\|^2,$$

and therefore

$$\frac{\mathrm{d}}{\mathrm{d}t}[f(x(t)) - f(x^*)] \leq -2\mu[f(x(t)) - f(x^*)].$$

Solving this differential equation leads to the desired result. We can apply a similar technique for the proof of (26), using that

$$\mu\|x - y\|^2 \leq \langle \nabla f(x) - \nabla f(y), x - y \rangle,$$

for strongly convex functions (see again Nesterov [2013]). ∎

## 12.2  Proof of Proposition 3.1

**Proposition.** *Given constants $0 < \mu \le L$, a step size $h > 0$ and a linear two-step method defined by $(\rho, \sigma)$, under the conditions*

$$
\begin{aligned}
(\rho_1 + \mu h \sigma_1)^2 &\le 4(\rho_0 + \mu h \sigma_0), \\
(\rho_1 + L h \sigma_1)^2 &\le 4(\rho_0 + L h \sigma_0),
\end{aligned}
$$

*the roots $r_\pm(\lambda)$ of $\pi_{\lambda h}$, defined in (8), are complex conjugate for any $\lambda \in [\mu, L]$. Moreover, the largest modulus root is equal to*

$$
\max_{\lambda \in [\mu, L]} |r_\pm(\lambda)|^2 = \max\{\rho_0 + \mu h \sigma_0, \ \rho_0 + L h \sigma_0\}.
$$

*Proof.* We begin by analyzing the roots $r_\pm$ of the generic polynomial

$$
z^2 + bz + c,
$$

where $b$ and $c$ are real numbers, corresponding to the coefficients of $\pi_{\lambda h}$, i.e. $b = \rho_1 + \lambda h \sigma_1$ and $c = \rho_0 + \lambda h \sigma_0$. If we want complex roots we need to satisfies

$$
b^2 \le 4c.
$$

After replacement, we need to satisfy for any $\lambda \in [\mu, L]$,

$$
b^2 - 4c \le 0 \quad \Leftrightarrow \quad (\rho_1 + \lambda h \sigma_1)^2 - 4(\rho_0 + \lambda h \sigma_0) \le 0.
$$

Since the left side this is a convex function in $\lambda$, it is equivalent to check only for the extreme values:

$$
\begin{aligned}
(\rho_1 + \mu h \sigma_1)^2 &\le 4(\rho_0 + \mu h \sigma_0), \\
(\rho_1 + L h \sigma_1)^2 &\le 4(\rho_0 + L h \sigma_0).
\end{aligned}
$$

In the complex conjugate case, the roots have the same modulus,

$$
|r_\pm(\lambda)|^2 = |c| = |\rho_0 + \lambda h \sigma_0|.
$$

Because the function is convex, the maximum is attained for an extreme value of $\lambda$,

$$
\max_{\lambda \in [\mu, L]} |r_\pm(\lambda)|^2 = \max\{\rho_0 + \mu h \sigma_0, \ \rho_0 + L h \sigma_0\},
$$

which is the desired result. ∎

## 12.3  Proof of Proposition 8.1

**Proposition.** *Let $f$ be $L$-smooth and convex and $x(t)$ be the solution of (Gradient Flow). Then*

$$
f(x(t)) - f(x^*) \le \frac{1}{\frac{t}{\|x_0 - x^*\|^2} + \frac{1}{f(x_0) - f(x^*)}} \le \frac{\|x_0 - x^*\|^2}{t + (2/L)}.
$$

*Proof.* Let $\mathcal{L}(x(t)) = f(x(t)) - f(x^*)$. We notice that $\nabla \mathcal{L}(x(t)) = \nabla f(x(t))$ and $\mathrm{d}\mathcal{L}(x(t))/\mathrm{d}t = -\|\nabla f(x(t))\|^2$. Since $f$ is convex,

$$
\begin{aligned}
\mathcal{L}(x(t)) &\le \langle \nabla f(x(t)), x(t) - x^* \rangle \\
&\le \|\nabla f(x(t))\| \|x(t) - x^*\|.
\end{aligned}
$$

By consequence,

$$
-\|\nabla f(x(t))\|^2 \le -\frac{\mathcal{L}(x(t))^2}{\|x(t) - x^*\|^2} \le -\frac{\mathcal{L}(x(t))^2}{\|x_0 - x^*\|^2}. \tag{27}
$$

The last inequality comes from the fact that $\|x(t) - x^*\|$ decreases over time,

$$
\begin{aligned}
\frac{\mathrm{d}}{\mathrm{d}t} \|x(t) - x^*\|^2 &= 2\langle \dot{x}(t), x(t) - x^* \rangle, \\
&= -2\langle \nabla f(x(t)), x(t) - x^* \rangle, \\
&\le 0 \quad \text{since } f \text{ is convex.}
\end{aligned}
$$

From (27), we deduce the differential inequality

$$\frac{\mathrm{d}}{\mathrm{d}t}\mathcal{L}(x(t)) \leq -\frac{\mathcal{L}(x(t))^2}{\|x_0 - x^*\|^2}.$$

The solution is obtained by integration,

$$\int_0^t \frac{\mathrm{d}\mathcal{L}(x(\tau))/\mathrm{d}\tau}{\mathcal{L}(x(\tau))^2}\,\mathrm{d}\tau \leq \int_0^t \frac{-1}{\|x_0 - x^*\|^2}.$$

The general solution is thus

$$\mathcal{L}(x(t)) \leq \frac{1}{\frac{t}{\|x_0 - x^*\|^2} + C},$$

for some constant $C$. Since the inequality is valid for all time $t \geq 0$, the following condition on $C$,

$$\mathcal{L}(x(t)) \leq \frac{1}{\frac{t}{\|x_0 - x^*\|^2} + C} \leq \frac{1}{C} \quad \text{for} \quad t \geq 0,$$

is sufficient. Setting $C = \frac{1}{f(x_0) - f(x^*)}$ satisfies the above inequality. If we use the fact that the function is smooth,

$$f(x(t)) - f(x^*) \leq \frac{L}{2}\|x_0 - x^*\|^2,$$

then we get the desired result. ∎