[Reviews · NeurIPS 2017]

Reviewer 1



The paper provides an interpretation of a number of accelerated gradient algorithms (and other convex optimization algorithms) based on the theory of numerical integration and numerical solution of ODEs. In particular, the paper focuses on the well-studied class of multi-step methods to interpret Nesterov's method and Polyak's heavy ball method as discretizations of the natural gradient flow dynamics. The authors argue that this interpretation is more beneficial than existing interpretations based on different dynamics (e.g. Krichene et al) because of the simplicity of the underlying ODE. Notice that the novelty here lies in the focus on multistep methods, as it was already well-known that accelerated algorithms can be obtained by appropriately applying Euler's discretization to certain dynamics (again, see Krichene et al). A large part of the paper is devoted to introducing important properties of numerical discretization schemes for ODEs, including consistency and zero-stability, and their instantiation in the case of multistep methods. Given this background, the authors proceed to show that for LINEAR ODES, i.e., for gradient flows deriving from the optimization of quadratic functions, natural choices of parameters for two-step methods yield Nesterov and Polyak's method. The interpretation of Nesterov's method is carried out both in the smooth and strongly convex case and in the smooth-only case. I found the paper to be an interesting read, as I already believed that the study of different numerical discretization methods is an important research direction to pursue in the design of more efficient optimization algorithms for specific problems. However, I need to make three critical points, two related to the paper in question and one more generally related to the idea of using multistep methods to define faster algorithms: 1) The interpretation proposed by the authors only works in explaining Nesterov's method for functions that are smooth in the l_2-norm and is not directly applicable for different norms. Indeed, it seems to me that different dynamics besides the gradient flow must be required in this case, as the method depends on a suitable choice of prox operator. The authors do not acknowledge this important limitation and make sweeping generalizations that need to be qualified, such as claiming that convergence rates of optimization algorithms are controlled by our ability to discretize the gradient flow equation. I believe that this incorrect when working with norms different from l_2. 2) The interpretation only applies to quadratic functions, which makes it unclear to me whether these techniques can be effectively deployed to produce new algorithms. For comparison, other interpretations of accelerated methods, such as Allen-Zhu and Orecchia's work led to improved approximation algorithms for certain classes of linear programs. 3) Regarding the general direction of research, an important caveat in considering multistep methods is that it is not completely clear how such techniques would help, even for restricted class of problems, for orders larger than 2 (i.e. larger than those that correspond to Nesterov 's AGD). Indeed, it is not hard to argue that both Nesterov's AGD (and also Nemirovski's Mirror Prox) can leverage smoothness to fully cancel the discretization error accrued in converting the continuous dynamics to a discrete-time algorithm. It is then not clear how a further reduction in the discretization error, if not properly related to problem parameters, could yield faster convergence. In conclusion, I liked the intepretation and I think it is worth disseminating it. However, I am not sure that it i significant enough in its explanatory power, novel enough in the techniques introduced (multistep methods are a classical tool in numerical systems) or promising enough to warrant acceptance in NIPS.